# Cost-aware orchestration of applications over heterogeneous clouds

**Kena Alexander**[1☯]**, Muhammad Hanif**[1☯]**, Choonhwa Lee**[1]*****, Eunsam Kim**[2]**, Sumi Helal**[3]

**1** Division of Computer Science and Engineering, Hanyang University, Seoul, Republic of Korea,
**2** Department of Computer Engineering, Hongik University, Seoul, Republic of Korea, **3** School of Computing and Communications, Lancaster University, Lancaster, United Kingdom

☯ These authors contributed equally to this work.
* lee@hanyang.ac.kr

**Data Availability Statement:** All relevant data are within the manuscript and its Supporting Information files. Note that simulated input data-set can be generated using our provided code backup.

## Abstract

The orchestration of applications and their components over heterogeneous clouds is recognized as being critical in solving the problem of vendor lock-in with regards to distributed and cloud computing. There have been recent strides made in the area of cloud application orchestration with emergence of the TOSCA standard being a definitive one. Although orchestration by itself provides a considerable amount of benefit to consumers of cloud computing services, it remains impractical without a compelling reason to ensure its utilization by cloud computing consumers. If there is no measurable benefit in using orchestration, then it is likely that clients may opt out of using it altogether. In this paper, we present an approach to cloud orchestration that aims to combine an orchestration model with a cost and policy model in order to allow for cost-aware application orchestration across heterogeneous clouds. Our approach takes into consideration the operating cost of the application on each provider, while performing a forward projection of the operating cost over a period of time to ensure that cost constraints remain unviolated. This allows us to leverage the existing state of the art with regards to orchestration and model-driven approaches as well as tie it to the operations of cloud clients in order to improve utility. Through this study, we were able to show that our approach was capable of providing not only scaling features but also orchestration features of application components distributed across heterogeneous cloud platforms.

## Introduction

One of the key concepts of the cloud computing paradigm is the reduction of cost of deploying and maintaining applications and components. Cloud portability and interoperability should allow for seamless use and reuse of components across various clouds at an acceptable cost [1]. While cloud computing concepts have provided solutions to key issues within the cloud such as vendor lock-in, there has only just been a slow increase in the work being done in the realm of cost analysis of cloud applications [2, 3].

**Funding:** This work was supported by the research fund of Hanyang University (HY-2019) and by the Basic Science Research Program through the National Research Foundation of Korea, funded by the Ministry of Science and ICT (Grant 2017R1A2B4010395 and Grant 2019R1A2C1002221). The funders had no role in study design, data collection and analysis, decision to publish, or preparation of the manuscript.

**Competing interests:** The authors have declared that no competing interests exist.

Recently, cloud application orchestration has gained considerable popularity as a means of addressing the much-needed cloud portability issue. Cloud orchestration allows applications to be easily distributed across heterogeneous cloud providers based on different metrics such as location and time. However, an important impact factor of cloud application orchestration remains the differences in the cost of the application and components, once they are distributed across heterogeneous cloud platforms. It may be possible to distribute applications across various cloud providers. However, if the cost impact of orchestration is not taken into consideration, it may be infeasible to do so [4, 5]. Therefore, there must be some standard methodology for determining the cost impact of orchestrating application and components across heterogeneous cloud providers.

Currently, TOSCA (Topology Orchestration Specification for Cloud Applications) [6] has established itself as a standard capable of defining the topology of an application to be deployed within the cloud. While TOSCA is a flexible specification that provides normative types that may be extended via new type definitions, the standard itself does not define how the concrete orchestration should be performed. As such, TOSCA does not define a methodology for cost analysis of cloud applications.

In this paper, we will present a cost-aware orchestration strategy that involves the combination of a topology orchestration standard TOSCA, a cloud application deployment and management platform CAMP [7], and a cloud application cost model capable of determining the operating cost of a cloud application. In our strategy, we intend to build upon the standard by adding features for cost analysis as well as include a proposed policy processing technique to orchestrate applications across heterogeneous cloud providers [8, 9]. The rest of this paper is structured as follows. In Section II, we present the motivating factors and challenges to our strategy. Section III introduces the architecture of our approach to cost-aware cloud orchestration. In Section IV, we present the main implementation concepts used to realize our proposed approach. Section V evaluates through experimentation and presents our results, after which we conclude our paper in Section VI.

## Motivation and challenges

In looking at the motivation for our study firstly, we considered the fact that a cloud provider provides virtual machine (VM) images based on different tiers, according to the hardware configuration of the servers required. As each tier of VM utilizes a varying amount of resources from the provider's infrastructure, it is only fitting that these tiers be priced differently based on the resources used in the provider's datacenter. Microsoft Azure, for example, categorizes its services based on use cases such as General Purpose, Compute Optimized, Memory Optimized, Storage Optimized, GPU and High Performance Compute [10]. Each of these is further subdivided. For example, General Purpose category is broken into B, A, and D tiers, as shown in Table 1. Amazon's AWS on the other hand provides a slightly different scheme for their VMs by dividing into both a kind (General, Compute, etc.) and a flavor. Table 2 gives us a look at a subsection of AWS's VM classification and division [11]. Apart from being subdivided based on the configuration of the VMs, cloud Service providers have also started providing VMs that are burstable such that their performance can burst above a given average and not incur added cost [12]. These VMs offer a baseline performance at a significantly lower cost to traditional VMs.

Importantly, what can be seen is that VMs are varied based on different metrics such as the number of cores and the amount of RAM that is available. Considering that an application may utilize varying numbers of VMs, each with different configurations, then the operating cost of the application should be related to the configuration of VMs used to deploy the

**Table 1. Microsoft Azure general purpose VM tier.**

| Instance | Cores | RAM | Temporary Storage | Price/hr |
|---|---|---|---|---|
| B-Series | | | | |
| B1S | 1 | 1 GB | 2 GB | $0.006 |
| B2S | 2 | 4 GB | 8 GB | $0.024 |
| B1MS | 1 | 2 GB | 4 GB | $0.012 |
| B2MS | 2 | 8 GB | 16 GB | $0.047 |
| B4MS | 4 | 16 GB | 32 GB | $0.094 |
| B8MS | 8 | 32 GB | 64 GB | $0.006 |
| A0 Basic | | | | |
| A0 | 1 | 0.75 GB | 20 GB | $0.018 |
| A1 | 1 | 1.75 GB | 40 GB | $0.024 |
| A2 | 2 | 3.50 GB | 60 GB | $0.068 |
| A3 | 4 | 7.0 GB | 120 Gb | $0.176 |
| A4 | 8 | 14.0 GB | 240 GB | $0.352 |
| Av2 Standard | | | | |
| A1 v2 | 1 | 2 GB | 10 GB | $0.036 |
| A2 v2 | 2 | 4 GB | 20 GB | $0.076 |
| A4 v2 | 4 | 8 GB | 40 GB | $0.159 |
| A8 v2 | 8 | 16 GB | 80 GB | $0.333 |
| A2m v2 | 2 | 16 GB | 20 GB | $0.099 |
| A4m v2 | 4 | 32 GB | 40 GB | $0.208 |
| A8m v2 | 8 | 64 GB | 80 GB | $0.437 |

application. Applications deployed to cloud service providers' infrastructure are usually subjected to varying conditions during their lifetime. Applications may be subjected to spikes in utilization, changes in the base cost of VMs used in their deployments or the addition or removal of flavors of VMs from the catalog of VMs offered by providers. As such any one of

**Table 2. Amazon AWS general purpose VM tier.**

| Kind | Flavor | CPU | RAM | Storage | Price |
|---|---|---|---|---|---|
| General | t2.small | 1 | 2 | EBS Only | $0.032 |
| General | t2.nano | 1 | 0.5 | EBS Only | $0.008 |
| General | t2.micro | 1 | 1 | EBS Only | $0.016 |
| General | m3.medium | 1 | 3.75 | 1x4 SSD | $0.096 |
| General | t2.medium | 2 | 4 | EBS Only | $0.064 |
| General | t2.large | 2 | 8 | EBS Only | $0.128 |
| General | m4.large | 2 | 8 | EBS Only | $0.129 |
| General | m3.large | 2 | 7.5 | 1x32 SSD | $0.193 |
| General | t2.xlarge | 4 | 16 | EBS Only | $0.256 |
| General | m4.xlarge | 4 | 16 | EBS Only | $0.258 |
| General | m3.xlarge | 4 | 15 | 2x40 SSD | $0.385 |
| General | t2.2xlarge | 8 | 32 | EBS Only | $0.512 |
| General | m4.2xlarge | 8 | 32 | EBS Only | $0.516 |
| General | m3.2xlarge | 8 | 30 | 2x80 SSD | $0.770 |
| General | m4.4xlarge | 16 | 64 | EBS Only | $1.032 |
| General | m4.10xlarge | 40 | 160 | EBS Only | $2.58 |
| General | m4.16xlarge | 64 | 256 | EBS Only | $4.128 |

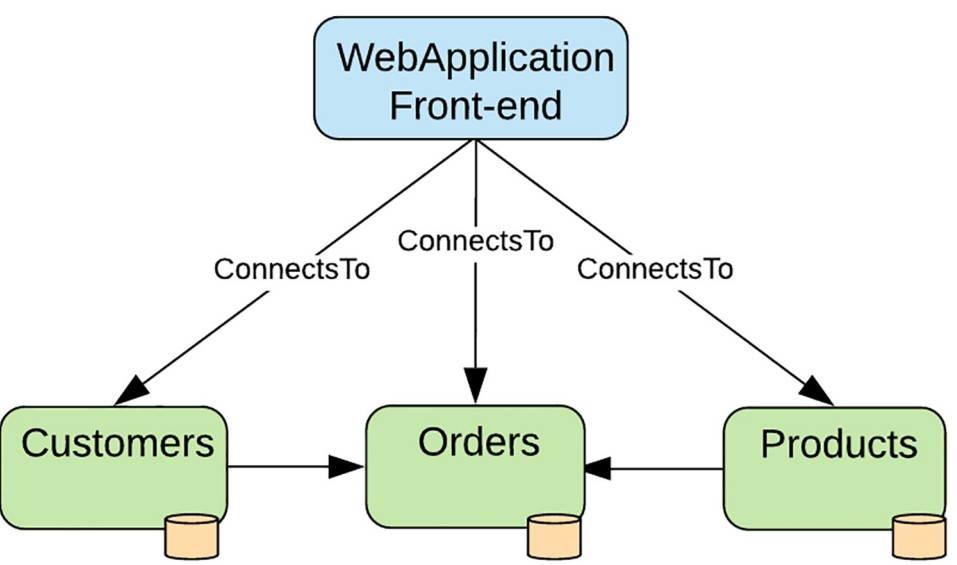

**Fig 1. Microservice-based application topology.**

these situations may result in changes in the cost to operate the application over time. It is therefore important that an application be aware of these changes and be capable of responding appropriately in order to mitigate their effects. Our work is, therefore, based on providing an orchestration strategy capable of managing the cost of an application by controlling the configuration of the VMs used to compose the application. To illustrate our strategy, we consider a situation involving a microservice application. The application comprises a front-end written in angular TS and running on Node.js as well as three back-end components, i.e., users, products, and orders. Each back-end component is a Java application that runs on an embedded Jetty server and uses its own Mongo DB data store. In the scenario illustrated in Fig 1, clients interact with the application via the application frontend. Customers create accounts on the system via the Customers microservice and the catalog of products sold by the company is handled by the Products microservice. When an order is made by a customer, the order system must accept the order and, by combining the customer information with the product information, the order can be fulfilled. A requirement of the system is that it should be deployed and managed within the cloud. Since the application is written using a microservices design, then it is possible that each service may be distributed across multiple cloud providers, when possible. Following deployment, it is thus the responsibility of the application operations team to manage the application in the cloud. If this application is required to operate within a specific budget, then the application operations team must monitor each component and determine that, as a whole, they do not exceed the budget. Apart from budgetary constraints, there may also a host of other factors that may affect the application such as changes in the cost model and pricing used by the cloud providers hosting the application. To allow for cost-aware application orchestration, we need to address the following key challenges.

## Topology model

In order to capture the components of the application, we need to express it in the form of a typed topological graph. By representing the application in such a form, we can identify each component of the application and consequently control the cost associated with it. To address this challenge, we represent the topology of the application using a TOSCA topological graph

[6]. TOSCA allows for the modeling of applications by describing the components using typed templates. These templates may be defined from normative/built-in types that are present in the specification or may be formed by extending the normative types to create new types.

## Cost model

Application blueprints deployed to the cloud may be fulfilled in different ways depending on the technology requested by the customer or employed by the vendor. For example, an application deployed to Kubernetes is fulfilled using Docker containers clustered to form logical pods, whereas an application deployed to Amazon's EC2 service may be fulfilled using virtual machine instances [13–15]. As it is possible for a service to be fulfilled through various means it is important that the model used to define the cost of an application be independent of vendor specific technology. Hence, the cost accrued by a service deployed on one provider remains unaffected in the event that that service must be fulfilled by another provider. Not only is it important that the model be vendor agnostic in order to allow cost to be accumulated regardless of the method used to fulfill the service, the model should also allow for the accumulation of cost for services deployed across multiple providers. This is usually the case with multi-cloud applications. To capture these properties, we chose to express the cost model as a function of the common factors of VMs used to fulfill services in provider clouds, then translate that function into an algorithmic form, embedding it into our policy orchestrator so that it may be combined with the topology and policies that define our application.

## Policy model

Apart from being able to model the cost constraints of an application deployed in the cloud, it is necessary to also enforce those constraints. In order to provide topology and cost modeling, we choose to use the TOSCA specification. TOSCA's specification from version 1.0 provided imperative policies in the form of BPMN or BPEL plans. These plans specify tasks that should be taken in the case that some event occurs. The state of the art has since moved to declarative policy specifications that define constraints of an application or components that should be adhered to. To address the need for a policy model, we also intend to provide a declarative policy specification through TOSCA.

Currently, TOSCA only specifies the model of the application. Therefore, it is necessary that we define a deployment strategy for the model. The deployment strategy used must also be capable of utilizing the declarative policies specified in TOSCA. For this, we make use of a CAMP platform that has been extended through the addition of a declarative policy specification. OASIS CAMP does not provide application orchestration features, nor does it have a specification that allows for policies. We, however, utilized an extended CAMP platform that allows for orchestration through declarative policy specification [9].

## Architectural design for cost-aware orchestration

Our approach to cost-aware application orchestration involves the combined use of the three models which can allow us to dynamically control a cloud application based on the cost of components. In previous studies, we demonstrated the combination of declarative standards of TOSCA and CAMP which were used for application delivery and management [8, 9]. In this work, we expand on that initial study by enhancing the combined model with a cost model that captures the cost information of an application. The overall architecture of our work, therefore, involves applications described in a TOSCA model that are translated into a CAMP model for deployment and management. The cost model information is then embedded into the TOSCA model through the use of properties, artifacts, and policies. The cost

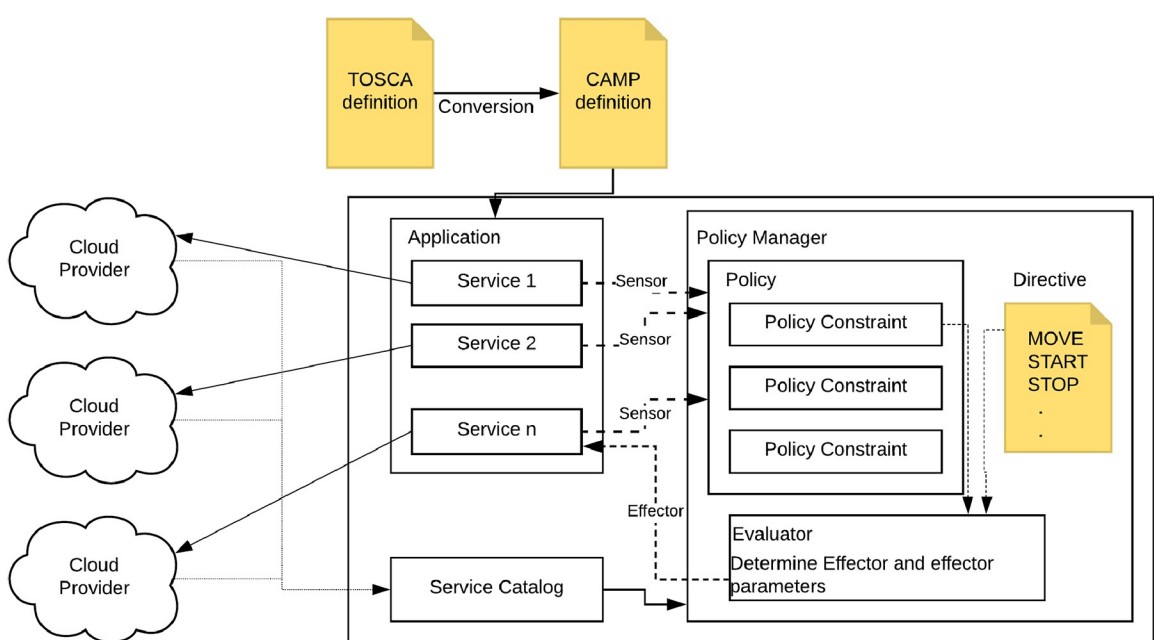

**Fig 2. Cost control orchestrator architecture.**

model is preserved through the TOSCA-to-CAMP model transformation process into our extended CAMP model ultimately as policy constraints and properties. We give a comprehensive overview of our approach in Fig 2 defining the details of the components as well as showing how they interact and interconnect. Here we show various cloud providers that are connected to our combined platform. TOSCA definitions are converted to CAMP and instantiated as an application comprising 1..n services. Each service interacts with our policy management component by exposing its sensor information as well as accepting effector actions. The policy management component is injected with a directive which defines possible actions allowed by the policy manager. Depending on whether or not sensor values exceed the policy constraints, the policy management component may initiate an effector by consulting its directives. The service catalog list services offered by various providers that may be consulted by the policy management component.

**TOSCA topology design.** As identified in an early section, the application topology model allows us to describe the components of the application using a provider-agnostic representation. For this representation, it is important that each component of the application be defined, such that the components that it is comprised of as well as their relationships are properly defined. For this, we make use of the Topology Orchestration Specification for Cloud Applications (TOSCA) [6]. TOSCA topology models allow application designers to model the overall structure of an application through the use of typed templates representing the application components. Each template is, in turn, referenced from a type definition which serves as a meta-model for the template. Therefore, when defining an application model, the components of the application must reference a defined application node template which in itself references a node type definition. To make use of TOSCA within our architecture, it is therefore necessary for us to define the node type definitions as well as the node templates that are capable of reflecting the information we need to capture about our application. We have thus defined additional, non-normative node type definitions, and making use of the attributes of those node definitions, we reflect the cost accrued for that component. This is shown in Fig 3. Here

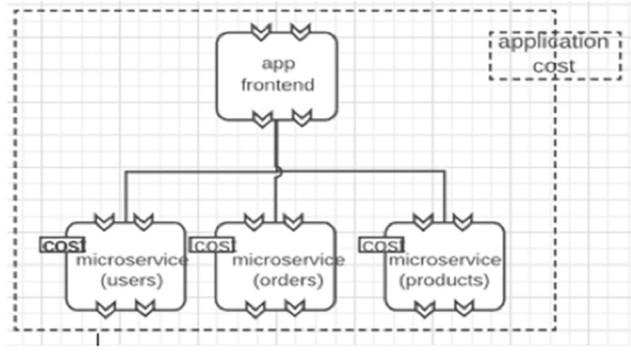

```
policies:
kr.ac.hanyang.nodes.microservices.Container:
    derived_from: tosca.nodes.Container.Runtime
    description: >
        The microservice Container Runtime node
    artifacts:
        deployment_file:
            type: tosca.artifacts.File
        attributes:
            operating_cost:
                type: float
                required: false
            fixed_cost:
                type: float
                required: false
        properties:
            port:
                ---
            data_port:
                ---
        capabilities:
            ---
    requirements:
    - microservice:
        ---
    - data_endpoint:
        ---
```

**Fig 3. TOSCA non-normative type showing cost attributes.**

we see an excerpt of the declarative types used to describe the services of the application. The node is of a container type and contains artifacts, attributes, properties capabilities and requirements. Each component of the service may hold information about the service or information used to instantiate the service on a provider and as can be seen, attribute information related to the operating cost and fixed cost of a component may be added.

Node templates in TOSCA specify the components of the application. However, in order to model the connection between these components, TOSCA provides relationship templates which are also represented as typed templates. Relationship templates specify how the application components should be connected to each other. Similar to node templates, relationship templates are defined from relationship type definitions and also specify attributes that refer to

the characteristics of the relationship between two nodes. For example, a software application node may be attached to another node via "ConnectsTo" relationship. As with node templates, relationship templates may also specify as attributes the cost accrued for the connection between the components. We take these into consideration, when defining our topology models, since it is possible to show the cost of components as well as the cost of connecting specific components together.

**Cost model for cloud orchestration.** The cloud computing model is a utility model whereby cloud providers deliver cloud services to clients at a cost. Cloud providers may fulfill application deliveries using different facilities. For example, IaaS and PaaS providers may fulfill application components through dedicated virtual machines upon which the client must install his software components. Depending on the approach that is used, clients may accrue a different cost over time. If we consider IaaS and PaaS platforms, applications of these platforms are comprised of artifacts that are owned by the client and deployed on the platform provided. If we were to consider each of these, then the cost of the application is influenced by the underlying infrastructure, making it possible to estimate the accrued cost of the application as well as determine a projected cost of the application over a period of time.

However, to determine the cost accrued by an application, it is necessary to develop and use a cost model that closely associates the cost of an application with its deployment configuration. We build our cost model on the fact that each component of the application can accrue cost based on the services used by the provider to fulfill it on their platform. Therefore, the application's cost should be the combination of the costs of each component that comprises the application. We illustrate our concept by first defining the rate as the amount charged per unit time to provision the application component within the provider's platform. We represent this as a tuple containing the factors that contribute to this rate, as can be seen in Eq 1. Secondly, we have defined the operating cost of an application component as the cost charged by the provider over a period of time t to provision and maintain the VM within its datacenter. The operating cost is, therefore, derived from the components' rates and the deployment life of the application components, as seen in Eq 2. Consequently, the operating cost of the application is the sum of the operating costs of each component as given in Eq 3.

$$\text{Rate} = C(l, i, c, m) \tag{1}$$

$$\text{Op}_{\text{comp}} = Rate * t \tag{2}$$

$$\text{Op}_{\text{app}} = \Sigma Op_{comp} \tag{3}$$

Where "l" represents location, "i" is instance/flavor, "c" is CPU, and "m" stands for memory.

**Policy design for cost-awareness.** Policy design is a critical aspect of our work, as policies form the back-bone of our application orchestration scheme. TOSCA service template documents provide the mechanism for specifying declarative policies that are enforced by the orchestrator engine during deployment.

**Typed policies.** Previously, we explored the use of declarative policies to control application deployment and orchestration [8, 9]. From this, we have seen that declarative policies are represented in TOSCA as typed components consisting of the constraints that are used to constrain the components of the application. In this work, we identify attribute extensions to node and relationship types. By leveraging these attributes extensions, through constraints, we have defined declarative policies that can be associated with the components of the application. Fig 4 gives an example of a declarative policy which states that an associated application should maintain an operating cost which is within the budgeted range.

```
policies:
- type: kr.ac.hanyang.oCamp.entities.policies.CostControl
    constraints:
    - properties: PROJECTED_OPERATING_COST
        type: kr.ac.hanyang.oCamp.entities.constraints.within_range
        value: [BUDGETED_COST]
    - properties: CPU_UTILIZATION
        type: kr.ac.hanyang.oCamp.entities.constraints.within_range
        value: [10, 60]
    targets: [webApplication]
```

**Fig 4. Policy minimizing application operating cost.**

Since declarative policies simply leverage the attributes defined by the application components, meaning is given to the policy only through its interpretation. We therefore incorporated the use of a policy manager which interprets the policies.

**Policy management.**    In our work, declarative policies represent a set of constraints that should be adhered to. However, our declarative policies do not specify actions that should be taken in the event of a violation. To capture this, we make use of another concept known as policy management directives that declaratively specify the properties that are expected to be altered by an action as well as the expected outcome of actions that may be taken in order to keep an application component in a valid state. For example, if a component violates a cost constraint, then the list of actions available to resolve the issue may include a "REFACTOR" action. Such an action states that, if a refactor is performed, then the operating cost of the component is expected to change from its initial value to a new value. An example of this can be seen in Fig 5. Here, if we consider a deployed application component, then that component will have properties representing the application's location (PROVISIONING_LOCATION) as well as the component's cost (SERVICE_COST). If the component's operating cost changes such that it violates the conditions of the policy, then the policy manager may reference the directives for an appropriate action to cause the property to change such that it is compliant again. The REFACTOR action seen here is shown to be capable of changing the SERVICE_COST of the component. The REFACTOR action, however, can also effect a change on other properties. Since these other properties are not part of our policy, the policy manager need not worry about their changes and may still choose to carry out a REFACTOR action. Conversely, a budget change may be indirectly caused by an increase in CPU performance of the VM. In this case, it is possible to affect the operating cost through the use of a REFACTOR or BURST_ABOVE action, if the current VM is burstable. It should be noted that policy directives serves to instruct how an attribute is expected to change, after an action is carried out. However, the algorithmic logic used in the selection of the directives is built into the policy manager.

## Proof-of-concept implementation

In this section, we present the implementation considerations made in our approach. Our approach aims to combine two standards TOSCA and CAMP through the use of a cost model in order to deliver a mechanism for providing cost-aware orchestration of application components deployed in the cloud.

**TOSCA-to-CAMP orchestrator.**    Our cost-aware orchestrator implementation platform makes use of extensions to work that was done prior involving the combination of TOSCA

```yaml
- id: REFACTOR
  actions:
  - property: PROVISIONING_LOCATION
    transitions:
    - type: Initial
      value: ANYTHING
    - type: Change
      value ANYTHING
  - property: SERVICE_COST
    transitions:
    - type: Initial
      value: ANYTHING
    - type: Change
      value ANYTHING
  - property: APPLICATION_COST
    transitions:
    - type: Initial
      value: ANYTHING
    - type: Change
      value ANYTHING
  - property: EXPECTED_APPLICATION_COST_PER_YEAR
    transitions:
    - type: Initial
      value: ANYTHING
    - type: Change
      value ANYTHING
- id: BURST_ABOVE
  actions:
  - property: CPU_CREDIT
    transitions:
    - type: Initial
      value: ANYTHING
    - type: DecreaseBy
      value: CPU_CREDIT_RATE"
```

**Fig 5. Refactor policy directive.**

and CAMP [9]. Our platform is composed of an in-house TOSCA parser capable of interpreting TOSCA topologies written in YAML and a CAMP platform that is used to deploy and manage the application. Our TOSCA parser parses the YAML file and an ATL model converter is used to convert the TOSCA model into a CAMP platform deployment plan. Using an extended CAMP platform, we were able to deploy the imported CAMP PDP (Platform Deployment Package) across heterogeneous cloud providers.

As the TOSCA standard is capable of making use of declarative policies, we extended our CAMP platform through the addition of a declarative policy specification, so that we may fully automate the orchestration of application end-to-end. Apart from the major components, our platform also relies heavily on another core component, i.e., our policy management engine, which we will discuss further in the following section, that is responsible for interpreting and processing the declarative policies used to manage the application.

**Cost-aware policy manager.** An important aspect of our work is the ability to properly determine the action required to resolve a cost constraint violation. For this, we use what we term as a policy management engine or policy manager for performing the analysis required. The policy manager registers policy management directives which contain actions that may be carried out in the event of a policy violation. As it is important for the policy manager to perform correctly in specific situations, the policy manager should be given proper directives. For example, if an application component is deployed using a burstable VM, then the policy manager should be aware of the option to burst above the baseline utilization in the event of a performance violation. Our policy manager is also afforded all the necessary information about the running application components, e.g., the location and tier of the deployed service as well as the configuration of the VM used to fulfil it. Consider an application component deployed using a VM based on Microsoft Azure's general purpose category using an A0 tiered VM. This image gives 1 CPU core and is priced at a rate of $0.018 per hour on demand as given in Table 1. The cost of 2 CPUs in this tier, however, jumps to $0.068 per hour. To make use of this information, our cost-aware policy manager utilizes a cost orchestration algorithm as well as a service catalog that allows it to query to the rate of services offered by different providers. Based on the actions that are available to the policy manager supplied as policy management directives, the policy manager uses our cost orchestration scheme to determine the actions to be carried out.

**Cost orchestration scheme.** Our cost orchestration algorithm is critical to determining the action as well as the carrying them out. Looking at our policy management directives, we have defined a REFACTOR action as an appropriate action to be carried out in the case of a violation of a cost constraint. Considering that a performance violation may also trigger a change in the application's topology, then it is also important that we define a BURST_ABOVE action as well. The definitions are shown in Fig 5. In the event that the policy manager decides that a REFACTOR or action is suitable to solve the issue, it is important that we properly evaluate them to ensure that when executed, results in a favorable solution. If we consider a scenario where we are required to reduce the operating cost of a service, we can identify the following actions:

1. Move (REFACTOR) the service to another VM on the same tier with a lower CPU core count.

2. Move (REFACTOR) the service to another VM on a different tier of the same provider, with or without lowering the CPU core count.

3. Move (REFACTOR) the service to another VM on another provider with or without lowering the CPU core count.

In the scenario where an increase in performance would necessitate a VM in a higher tier which would incur a higher cost then in order to maintain the current operating cost, the following actions may be considered.

1. Increase (BURST_ABOVE) the CPU performance of the current VM to meet the performance needs rather than triggering a change of VM.

```
public void refactorDown(BUDGET) {
    exit = false;
    while(!exit){
        if (application.violation(BUDGET)){
            refactored = false;
            sortedServices = sortDescending(services);
            for each Service svc in sortedServices {
                current = getCurrentProvider(svc);
                refactored = (!performRefactor(svc).equalsTo(current) || false );
                if (!application.violates(BUDGET)) {
                    exit = true;
                }
            }
            if ( !refactored ) exit = true;
        } else
            exit = true
    }
}

public void refactorUp(BUDGET){
    exit = false;
    while(!exit){
        if (application.violation(BUDGET)){
            refactored = false;
            sortedServices = sortAscending(services);
            for each Service svc in sortedServices {
                current = getCurrentProvider(svc);
                refactored = (!performRefactor(svc).equalsTo(current) || false );
                if (!application.violates(BUDGET)) {
                    exit = true;
                }
            }
            if ( !refactored ) exit = true;
        } else
            exit = true
    }
}

performRefactor(Service svc, Catalog stream) {
    Provider min1 = stream.filter(svc.location, svc.tier)
    .min(Comparator.comparing(Service::getRate()))
    .orElse(DEF_RATE);
    Provider min2 = stream.filter(svc.location, svc.tier)
    .min(Comparator.comparing(Service::getRate()))
    .orElse(DEF_RATE);
    Provider min2 = stream.filter(svc.location, svc.tier)
    .min(Comparator.comparing(Service::getRate()))
    .orElse(DEF_RATE);

    return MinProvider(min1, min2, min3)
}

performBurst(Service svc) {
    final ScheduledExecutorService executorService =
    Executors.newSingleThreadScheduledExecutor();
    executorService.scheduleAtFixedRate(new Runnable() {
        @Override
        public void run() {
            if (svc.cpuCredit() <= 0 || svc.cpuUtilization() <= svc.BASELINE) return;
        }
    }, 0, 1, TimeUnit.SECONDS);
}
```

**Fig 6. Cost orchestration algorithm.**

Our policy manager must therefore be equipped with an algorithm to determine how to move the service in order to mitigate the cost constraint violation. The outline of the algorithm used in our work is given in Fig 6. Here we used a greedy algorithm to test the responses of the responses of the policy manager to ensure that it evaluated the directives it was given. The algorithm contains modules for refactoring as well as bursting. Depending on the scenario, either may be chosen as a viable option to rectify a policy violation. The module will loop until the service is refactored or it exits due to there being no viable option to refactor to. The refactor sub-module finds three minimum value services that may resolve the violation: the smallest service on the same tier on the same provider, the smallest service of another tier on the same provider, and the smallest service of any tier on another provider. The minimum of these three is the smallest service that may resolve the violation. In the case of a performance violation, a

burst action will always be considered over performing a refactor action, as this action does not alter the operating cost.

## Evaluation of results

In evaluating our work we chose to perform a policy enforcement performance evaluation and a cost-awareness evaluation. For our policy enforcement performance evaluation, it was important that we measure key areas of performance of our approach. For this we chose a two point approach,

- Determine the time to detect a cost violation

- Determine how our approach scales with multiple services

In determining the time it takes to detect and initiate an action for a cost violation, it was clear that our approach needed to be able to detect a violation within the smallest billing period used by cloud service providers. A majority of cloud service providers make use of per second billing, hence, it was important that we determine whether our approach can detect a violation in time less than one (1) second. Once we established if our approach can feasibly detect a violation within an appropriate time, it was important that we determine how our approach scales to applications comprising more than one node. As most microservice applications are represented by multiple, interconnected service, then our approach needed to scale such that applications of more than one node also provided detection times of less than one (1) second. For our cost-awareness evaluation we chose to determine how our approach reacts to various scenarios. It was important that our approach be capable of not only detecting a violation but also determining the appropriate action in each scenario. The testbed used for our evaluation comprises our test microservice application as presented in Fig 1 on a MacBook Pro 2016 running with a 2.6 GHz Intel i7 CPU and 16GB of RAM. We supplied our TOSCA and CAMP platforms with the TOSCA application plans and policy management directives respectively, performed our model conversion between the TOSCA application topology and the CAMP application deployment plan. We then fed simulation data into our orchestrator and observed the responses to the scenarios presented.

### Policy enforcement performance

First, we need to determine the time taken to evaluate a REFACTOR or BURST_ABOVE decision for a single service, when its cost constraint gets violated. To do this, we created a deployment with a single service, triggered a cost violation on that service, and recorded the time required to evaluate the decision. The results were tabulated and the experiment was then repeated. The tests were run for approximately 5000 iterations, after which we were able to determine the average time to detect and respond to a cost constraint violation. This was found to be approximately 0.9846ms. Fig 7 shows the results of this baseline test.

We then increased the number of services in our test application and ran the experiment again, this time recording the time taken to evaluate the refactor actions for two service, three services, and up to an application with ten service nodes. In these cases, we however reduced the number of iterations that were run in an effort to obtain results in a timely manner. The results of these extended tests were tabulated and a graph of the average detection time per node was obtained. We were able to deduce that the time taken to evaluate a decision by our policy manager was relatively constant regardless of the number of nodes present. These results are given in Fig 8.

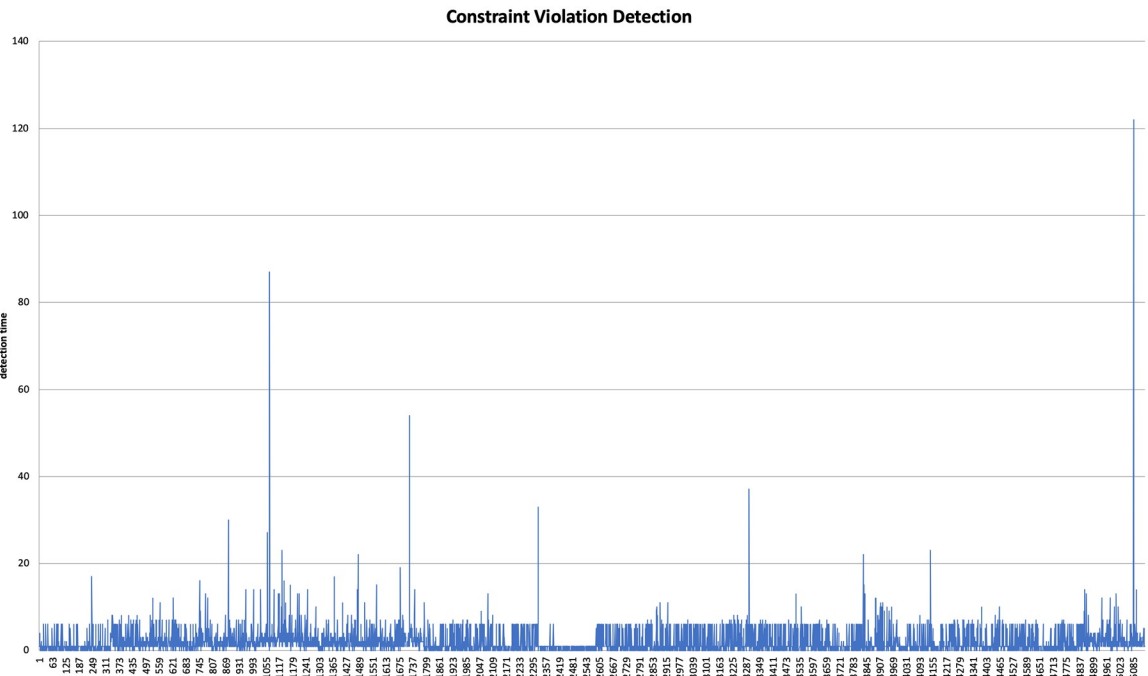

**Fig 7. Baseline conversion times.**

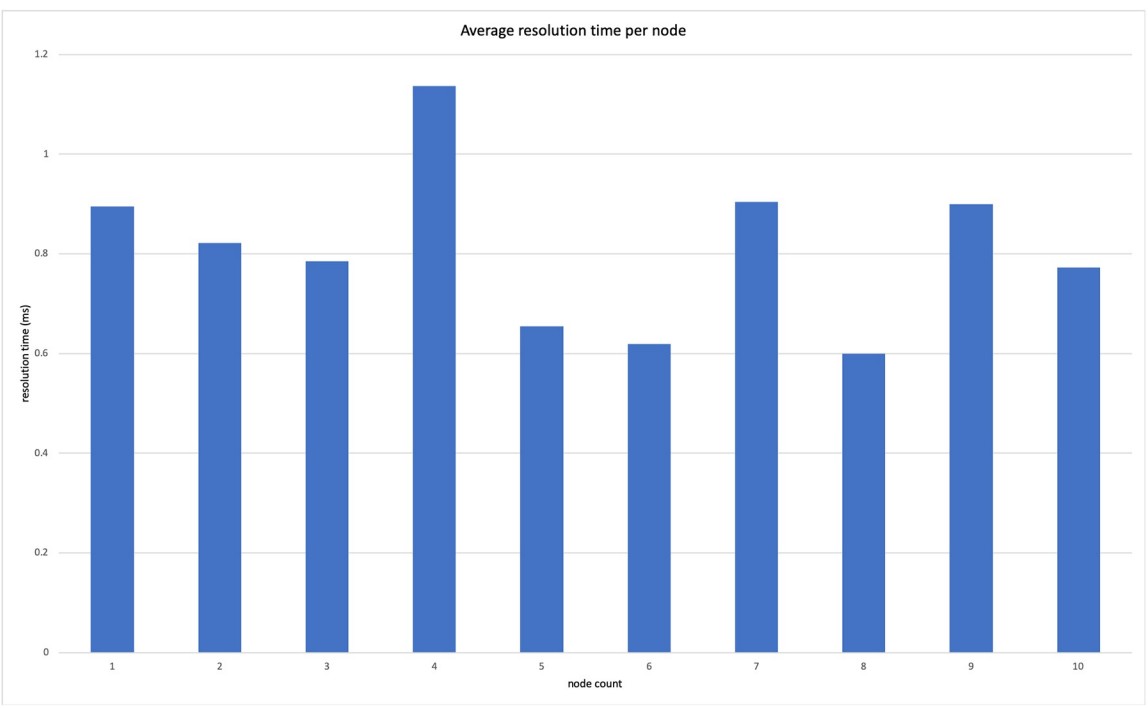

**Fig 8. Average resolution time per node.**

## Cost-awareness evaluation

To validate our cost-aware orchestration approach, we refer to the example application that was presented in Fig 1 as a test microservice application. We decided to run live tests of this application deployed across different provider clouds and using various scenarios. Before we proceed, however, we will further explain the policy evaluation process used by our approach. Within our TOSCA and CAMP platform, our policy engine is composed of various policy managers, each capable of adequately resolving a policy violation based on the type of the policy that was specified. For example, a policy of type of __ kr.ac.hanyang.oCamp.entities.policies. CostControl can be efficiently handled by a Cost Control policy manager, while other policy types are handled by their corresponding policy manager types. Each policy manager type is therefore equipped with the algorithms allowing it to manage a specific scenario. For our experiment, we subjected the system to different scenarios and then recorded the response of the system for each. In our scenarios, we treat budgeted cost as a range allowing us to manipulate the lower and upper bounds of the range. The scenarios explored are as follows:

- Response to increase in upper bounds of Budget.

- Response to decrease in upper bounds of Budget.

- Response to increase in lower bounds of Budget.

- Response to decrease in lower bounds of Budget.

- Response to Increase in CPU utilization above Baseline.

Prior to beginning the tests, we configured the application with an initial cost control policy which specifies that our application's cost should be maintained between the range of $300 and $500. Once specified, we deployed the application components across the cloud providers. The initial component distribution and configuration is given in Table 3. We then ran each scenario and tabulated the results obtained from our system which we present in Table 4. We can see that, in the first scenario, if the upper limit of the budget is increased to $600, then the system responds with no change. In scenario two, the upper budget constraint is decreased to $350, which caused the system to respond by refactoring comp 1 and comp 2. The refactor actions resulted in the service being moved to VMs of the same flavours in the a different region. When the lower budget was reduced, however, no refactoring was recorded. Also, no refactoring was done for scenarios 1 and 4, as these scenarios would not result in a policy violation. Finally in the last scenario, when the cpu utilization was increased, rather than performing a refactor to another VM, the policy manager instead performed a burst action which cause the budget to remain unchanged.

**Table 3. Initial component distribution.**

| **Initial** | | | | | |
|---|---|---|---|---|---|
| | l | I | c | m | Rate |
| comp 1 | AWS Tokyo | t2.large | 2 | 8 | $0.128 |
| comp 2 | AWS Sydney | t2.large | 2 | 8 | $0.128 |
| comp 3 | Rackspace HK | General 1-4 | 2 | 3.75 | $0.148 |
| comp 4 | Rackspace SYD | General 1-4 | 2 | 3.75 | $0.148 |
| | | Projected Operating Cost | | | $397.440 |

**Table 4. Response to budget constrain changes and CPU utilization.**

| Scenario | | Action | Final | l | i | c | m | Rate |
|---|---|---|---|---|---|---|---|---|
| 1. Upper Budget Increase | | No Action | comp 1 | AWS Tokyo | t2.large | 2 | 8 | $0.13 |
| $300.00 | $600.00 | No Action | comp 2 | AWS Sydney | t2.large | 2 | 8 | $0.13 |
| | | No Action | comp 3 | Rackspace HK | General 1-4 | 4 | 4 | $0.15 |
| | | No Action | comp 4 | Rackspace SYD | General 1-4 | 4 | 4 | $0.15 |
| | | | | | Projected Operating Cost | | | $397.44 |
| 2. Upper Budget decrease | | Refactor | comp 1 | AWS Virginia | t2.large | 2 | 8 | 0.094 |
| $300.00 | $350.00 | Refactor | comp 2 | AWS Virginia | t2.large | 2 | 8 | 0.094 |
| | | No Action | comp 3 | Rackspace HK | General 1-4 | 4 | 4 | 0.148 |
| | | No Action | comp 4 | Rackspace SYD | General 1-4 | 4 | 4 | 0.148 |
| | | | | | Projected Operating Cost | | | $348.48 |
| 3. Lower Budget Increase | | Refactor | comp 1 | AWS Tokyo | m3.large | 2 | 7.5 | $0.19 |
| $450.00 | $500.00 | Refactor | comp 2 | AWS Sydney | m3.large | 2 | 7.5 | $0.19 |
| | | No Action | comp 3 | Rackspace HK | General 1-4 | 4 | 4 | $0.15 |
| | | No Action | comp 4 | Rackspace SYD | General 1-4 | 4 | 4 | $0.15 |
| | | | | | Projected Operating Cost | | | $486.00 |
| 4. Lower Budget decrease | | No Action | comp 1 | AWS Tokyo | t2.large | 2 | 8 | $0.13 |
| $200.00 | $500.00 | No Action | comp 2 | AWS Sydney | t2.large | 2 | 8 | $0.13 |
| | | No Action | comp 3 | Rackspace HK | General 1-4 | 4 | 4 | $0.15 |
| | | No Action | comp 4 | Rackspace SYD | General 1-4 | 4 | 4 | $0.15 |
| | | | | | Projected Operating Cost | | | $397.44 |
| 5. Increase CPU utilization | | | | | | | | |
| 15% | 65% | Burst_Above | comp 1 | AWS Tokyo | t2.large | 2 | 8 | $0.13 |
| 15% | 65% | Burst_Above | comp 2 | AWS Sydney | t2.large | 2 | 8 | $0.13 |
| | | No Action | comp 3 | Rackspace HK | General 1-4 | 4 | 4 | $0.15 |
| | | No Action | comp 4 | Rackspace SYD | General 1-4 | 4 | 4 | $0.15 |
| | | | | | Projected Operating Cost | | | $397.44 |

## Qualitative comparison of alternative techniques

The analysis and comparison of the cost of cloud computing is considered a fundamental function in order to mitigate some of the issues inherent in cloud computing, vendor lock-in being among the more prevalent ones. There have been other approaches aiming to provide this function to the consumers of cloud computing. With our approach being amongst them, it is therefore fitting that we provide a qualitative comparison of the approaches in order to show exactly where our approach fits into the landscape of cloud computing cost control.

In creating our comparison, we classified the ability of each solution to provide some key features of cost control, as it pertains to cloud computing. These include:

- Cloud service monitoring—The ability of the solution to provide cost monitoring for solutions deployed on a cloud platform.

- Cloud service cost optimization The ability of the solution to optimize the cost of solutions deployed in the cloud as well as predict cost based on current utilization.

- Multi-cloud support—The ability of the solution to provide cost control for services deployed across multiple clouds.

- Use of standards—Whether the solution makes use of cloud computing standards as part of their solution.

**Table 5. Qualitative comparison of cost control solutions.**

| | | Cloud Chekr | Right Scale | Cloud Health Technologies | Our Solution |
|---|---|---|---|---|---|
| Cloud Service Monitoring | Per service monitoring | Provides per service monitoring for infrastructure service deployed in supporting provider platforms. | Provides per service monitoring for infrastructure service deployed in supported provider platform | Provides per service monitoring for Infrastructure service deployed in supporting provider platforms. | Provides per service monitoring for infrastructure service deployed in supporting provider platforms. Provide at the application level. |
| | Application monitoring | Infrastructure services are not consolidated into applications. | Infrastructure services are not consolidated into applications. | Infrastructure services are not consolidated into applications. | Infrastructure services are not consolidated into applications. |
| Cloud Service Cost Optimization | Reactive Cost Optimization | Provides usage reports of service utilization per provider. Reports can be used to keep or discard deployed service. | Provides usage reports of service utilization per provider. Reports can be used to keep or discard deployed services. | Provides usage reports of service utilization per provider. Reports can be used to keep or discard deployed services. | Can provide usage reports of service utilization per provider. |
| | Proactive Cost Optimization | Provides proactive recommendations based on desired savings. | Provides proactive recommendations based on desired savings. Provides a policy engine that provides cost forecasting. | Provides proactive recommendations based on desired savings. Provides a policy engine that provides cost forecasting. | Can proactively determine recommendations based on budget. Provides a policy engine that is capable of forecasting. (Depends on the policy manager and algorithm used) |
| Multi-cloud support | Multiple Cloud platforms support | Provides support for different Cloud providers. | Provides support for different cloud providers. | Provides support for different cloud providers. | Provides support for different Cloud providers. |
| | Cost consolidation across multiple providers | Allows generation of reports of service deployed on different providers. | Allows generation of reports of service deployed on different providers. Provides scheduling and lifecycle. | Allows generation of reports of service deployed one different providers. Provides scheduling and lifecycle. | Allows generation of reports of service deployed on different providers. Provides scheduling and lifecycle. |
| | Multi-cloud orchestration of application components | Does not orchestrate cloud optimization policies across various providers. | Does not orchestrate cloud optimization policies across various providers. | Does not orchestrate cloud optimization policies across various providers. | Can orchestrate cloud optimization policies across various providers. |
| Use of standards | Open standards or proprietary technology | Based on proprietary technology. | Based on proprietary technology. | Based on proprietary technology. | Based on open technology TOSCA and CAMP. |

We present the results of our analysis within Table 5. Here we see that, with regards to cloud service monitoring, each measured solution in question, while capable of providing per service (component-aware) monitoring, remains unaware of the interconnection of the services as an application. When it comes to optimization of cost, all are capable of reactive reporting. However, some lack the ability to provide proactive optimization by forecasting. Support for multiple cloud platforms is also provided by all of the measured solutions. However, cost consolidation across different providers as well as the ability for orchestration policies to span providers were not found in most. Finally, with regards to the use of standards, all measured solutions were found to be proprietary except for our solution.

## Related work

Cloud application orchestration is a fairly young area of distributed and cloud computing. However, there has been a considerable amount of traction in the field [16–18], usually taking on either a declarative or imperative approach [19–21]. With an imperative approach, an application orchestration engineer is usually required to determine the conditions that should trigger an orchestration event as well as provide scripts that should be able to handle those situations as well as unforeseen ones. With TOSCA v1.1 [6], declarative workflows and triggers were added to the specification in order to provide alternatives to the imperative approach that prevailed. The workflows define steps that may be carried out in parallel or sequentially in

response to an orchestration event. However, it should be noted that TOSCA does not make considerations for cost control, so that its normative components do not contain properties and attributes to hold cost information. The imperative approach with TOSCA will also require imperative rules to be written for each possible situation that may arise. This may seem infeasible, when considering the number of nodes in an application as well as the situations that may trigger different orchestration responses.

Another alternative approach to providing cost control to cloud application involves the use of external services that allow for tracking and, to an extent, controlling the cost of applications across cloud platforms. Services such as Cloud Health Technologies, RightScale, and CloudCheckr [22–24] allow for customers to plan their cloud application deployment by providing detailed information about the services and best practices available from each providers to benefit key decision makers. The typical process of these approaches involves first determining the services required as well as the cost of these services using the library provided by the service providers. The application designer can then budget the application's utilization and ensure that the services required are within budget. Our work aims to extend further than these approaches by providing an automated process, based on an emerging standard, for analyzing service cost and orchestrating application components across service providers.

In addition to the aforementioned approaches, there have been various work dedicated to cost-aware scaling of cloud applications [25–28]. A cost-aware system, named Kingfisher solution was presented that considers other elasticity mechanisms such as replication and migration [25]. Another group made use of a linear regression model to predict the workload of an application which can then be used to determine scaling [26]. By combining the workload prediction model with a cost minimization approach, they minimized cost, while providing scaling. On the other hand, it was shown that combining the use of a cost-aware scaling model and a workload-adaptive scaling model provided effective scaling criteria for their scaling approach [27]. These works take a cost-aware approach to cloud computing by focusing mainly on auto-scaling rather than multi-cloud orchestration of each component of the application, which is a primary concern in our case.

Finally, there have been considerable work done on the analysis of the performance of application topologies as well as algorithms focused on cost constraint control [29, 30]. These works show that it is not only important to consider the cost-related factor of a distributed application, but also other factors should be considered as well, since application performance plays an important part in determining the final deployment of an application in the cloud. For instance, it was shown that it is possible and even necessary to determine the performance of an deployed application [29]. Also, it was argued that for workflows described as directed acyclic graphs, there are efficient methods for managing performance under cost constraints [30]. We acknowledge the importance of these approaches and appreciate the possible applications within our study. However, for our approach, we decided that through the use of a cost model defined as a tuple combining key differentiators, we can filter service offerings provided via a service catalog. By combining this with our declarative policy model, our approach can not only scale within a provider's platform, but also orchestrate components across providers. We see this as not just an incremental extension of the work done but also an adoption of a new, novel approach to cost control orchestration through the addition of our policy management capabilities which simplifies the process of determining and implementing cost control measures. Our approach is found to be capable of matching the features of controlling cost on a per-component basis. Furthermore, it is also capable of providing application-aware cost control as opposed to other cost optimization schemes approaches that only incorporate component-level scaling.

## Conclusion

In this paper, we presented our approach to cost-aware application orchestration across heterogeneous platforms. For this, we proposed an approach that incorporated the use of the standards, TOSCA and CAMP for modeling and deploying the application in the cloud, coupled with a cost model that is capable of capturing the factors that influence the cost of an application component. We coupled these to our declarative policy specification which allows us to orchestrate application components based on cost constraints. We demonstrated the ability to analyze the cost data of each provider by aggregating the data through a service catalog and the combined use of a cost control algorithm within our policy support mechanism. We were also able to demonstrate through a proof of concept that our approach is capable of providing cost-aware cloud orchestration of an application within a real-world situation. Our approach was found to be capable of providing the advantages of application-aware cost control capabilities as well as matching the component-level cost control features of other solutions.

## Supporting information

**S1 File. Raw-Results: The raw data from the evaluation of the proposed system.** This file contains the raw experimental results from the evaluation of the proposed system.
(XLSX)

**S2 File. Code-Backup: The implementation of the proposed system.** This zip file contains the implementation of the proposed system and its related files.
(ZIP)

## Author Contributions

**Conceptualization:** Kena Alexander, Muhammad Hanif, Choonhwa Lee, Sumi Helal.

**Funding acquisition:** Choonhwa Lee.

**Investigation:** Muhammad Hanif.

**Methodology:** Kena Alexander, Choonhwa Lee, Sumi Helal.

**Software:** Kena Alexander.

**Supervision:** Choonhwa Lee, Sumi Helal.

**Validation:** Kena Alexander, Choonhwa Lee, Eunsam Kim.

**Visualization:** Muhammad Hanif, Eunsam Kim.

**Writing – original draft:** Kena Alexander, Muhammad Hanif.

**Writing – review & editing:** Choonhwa Lee, Eunsam Kim.

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
