## [Decision Letter · Decision Letter 0]

16 Oct 2019

PONE-D-19-20789

Cost-Aware Orchestration of Applications over Heterogeneous Clouds

PLOS ONE

Dear Dr. Lee,

Thank you for submitting your manuscript to PLOS ONE. After careful consideration, we feel that it has merit but does not fully meet PLOS ONE’s publication criteria as it currently stands. Therefore, we invite you to submit a revised version of the manuscript that addresses the points raised during the review process.

Specifically, as the reviewers have highlighted, there are major concerns related to the lack of a proper case for the proposed work, the inadequacy of the details about the proposed scheme, and the weak presentation of the text and figures in the paper. The reviewers have also felt that the novelty of the work is weak and the authors should improve the novelty of the work. For example, the authors could improve the novelty by incorporating dynamic pricing in their work or give an explanation as to why it has not been included.

We would appreciate receiving your revised manuscript by Nov 30 2019 11:59PM. To enhance the reproducibility of your results, we recommend that if applicable you deposit your laboratory protocols in protocols.io, where a protocol can be assigned its own identifier (DOI) such that it can be cited independently in the future. For instructions see: http://journals.plos.org/plosone/s/submission-guidelines#loc-laboratory-protocols

We look forward to receiving your revised manuscript.

Kind regards,

Rashid Mehmood, PhD

Academic Editor

PLOS ONE

Journal Requirements:

2. Thank you very much for your submission to PLOS ONE. Before we proceed, we kindly ask that you address the following:

*Please explain the rationale for the development of your method in light of recent research in this area, clearly indicating which problem with existing methods you are addressing.

*Please clearly report at the beginning of your methods or results section which were the key performance measures used to establish the validity and utility of your method. Please also report clearly which statistical analysis was used to establish robustness of performance measures.

* Please note that PLOS ONE requires that experiments, statistics, and other analyses must be performed to a high technical standard and described in sufficient detail to allow for reproducibility of the study (http://journals.plos.org/plosone/s/criteria-for-publication#loc-3). To demonstrate the performance of the method, we would expect comparisons to be drawn between existing state-of-the-art methods.

*Were any deal datasets used to test the model? If so, please ensure that these are clearly listed in the beginning of your Methods section, as well as within your Data availability statement, in enough detail for another researcher to obtain the same datasets and be able to reproduce the findings.

Thank you for your attention to these queries.

Reviewers' comments:

Reviewer's Responses to Questions

**Comments to the Author**

1. Is the manuscript technically sound, and do the data support the conclusions?

Reviewer #1: Yes

Reviewer #2: Yes

2. Has the statistical analysis been performed appropriately and rigorously? 

Reviewer #1: N/A

Reviewer #2: Yes

3. Have the authors made all data underlying the findings in their manuscript fully available?

Reviewer #1: No

Reviewer #2: Yes

4. Is the manuscript presented in an intelligible fashion and written in standard English?

Reviewer #1: Yes

Reviewer #2: Yes

5. Review Comments to the Author

Reviewer #1: Thanks for submitting your work! The paper overall is well written and describes a way to do cost-aware orchestration on a application component level. This is an important part of orchestration and worthy to be published.

The results look promising, albeit a bit incremental to related work - as also pointed out by the authors in the related work section. However I belief that with some minor revisions this work could be set apart from what has been done to date and show a greater impact. I'd suggest to look at dynamic pricing e.g. burstable instances in public Clouds like AWS. For example by knowing that some components of the application have low average CPU utilization but have occasionally utilization spikes - further cost saving could be achieved. This dynamicity would be a nice addition to the work presented in the paper - For initial reference see: https://docs.aws.amazon.com/AWSEC2/latest/UserGuide/burstable-performance-instances-unlimited-mode.html

Hope the authors consider this recommendation - and I wish them the best for there submission!

Reviewer #2: This work focused on the orchestration strategy to manage the cost of applications over clouds. The authors considered a scenario of microservice applications to illustrate their approach. After reviewing this paper, couples of comments are provided below.

1) In the motivation and challenges part, The authors illustrated the motivation to control the cost of an application by different configurations of VMs. But the importance of the cost-aware orchestration has not been addressed. Before talking about challenges in the topology model, cost model, and policy model respectively, the relationship among these three models should be explained clearly. And then what the problems and difficulties in designing can be illustrated.

2) In the Cost Model for Cloud Orchestration, the model is defined in the component level. The reason for the cost model is independent of cloud providers platforms should be explained. And which parts make your approach application-aware need to be clarified.

3) The authors did a good study in the related work for TOSCA and other external cost control services. A minor error should be noted. ...[?, 20, 26]... in line #427.

4) Figure 4,5,6 and table 1,2 look like a screenshot. Please try to use latex or other utilities to generate table or algorithm blocks to make them clear enough when zooming in.

6. PLOS authors have the option to publish the peer review history of their article (what does this mean?). If published, this will include your full peer review and any attached files.

Reviewer #1: No

Reviewer #2: No

---

## [Author Response · Author response to Decision Letter 0]

15 Dec 2019

Please find the response letter attached.

---

## [Decision Letter · Decision Letter 1]

8 Jan 2020

Cost-Aware Orchestration of Applications over Heterogeneous Clouds

PONE-D-19-20789R1

Dear Dr. Lee,

We are pleased to inform you that your manuscript has been judged scientifically suitable for publication and will be formally accepted for publication once it complies with all outstanding technical requirements.

With kind regards,

Rashid Mehmood, PhD

Academic Editor

PLOS ONE

Additional Editor Comments (optional):

Reviewers' comments:

Reviewer's Responses to Questions

**Comments to the Author**

1. If the authors have adequately addressed your comments raised in a previous round of review and you feel that this manuscript is now acceptable for publication, you may indicate that here to bypass the “Comments to the Author” section, enter your conflict of interest statement in the “Confidential to Editor” section, and submit your "Accept" recommendation.

Reviewer #1: All comments have been addressed

2. Is the manuscript technically sound, and do the data support the conclusions?

Reviewer #1: Yes

3. Has the statistical analysis been performed appropriately and rigorously? 

Reviewer #1: Yes

4. Have the authors made all data underlying the findings in their manuscript fully available?

Reviewer #1: Yes

5. Is the manuscript presented in an intelligible fashion and written in standard English?

Reviewer #1: Yes

6. Review Comments to the Author

Reviewer #1: Thanks for considering the comments of the reviewers; Happy to see the additions made, especially to capture the bursting capabilities. I would like to encourage the authors to do further research in this space and publish in future!

7. PLOS authors have the option to publish the peer review history of their article (what does this mean?). If published, this will include your full peer review and any attached files.

Reviewer #1: No

---

## [Editor Report · Acceptance letter]

31 Jan 2020

PONE-D-19-20789R1 

Cost-Aware Orchestration of Applications over Heterogeneous Clouds 

Dear Dr. Lee:

I am pleased to inform you that your manuscript has been deemed suitable for publication in PLOS ONE. Congratulations! Your manuscript is now with our production department. 

With kind regards,

on behalf of

Dr. Rashid Mehmood 

Academic Editor

PLOS ONE